# Potential Benefits of Dietary Plant Compounds on Normal and Tumor Brain Cells in Humans: In Silico and In Vitro Approaches

**DOI:** 10.3390/ijms24087404

**Published:** 2023-04-17

**Authors:** Lucia Camelia Pirvu, Georgeta Neagu, Adrian Albulescu, Amalia Stefaniu, Lucia Pintilie

**Affiliations:** 1Department of Pharmaceutical Biotechnologies, National Institute of Chemical Pharmaceutical Research and Development—ICCF, 112 Vitan Av., 031299 Bucharest, Romania; director-general@ncpri.ro; 2Department of Pharmacology, National Institute of Chemical Pharmaceutical Research and Development—ICCF, 112 Vitan Av., 031299 Bucharest, Romania; getabios@yahoo.com (G.N.); or rockady@gmail.com (A.A.); 3Stefan S. Nicolau Institute of Virology, 285 Mihai Bravu Av., 030304 Bucharest, Romania; 4Department of Synthesis of Bioactive Substances and Pharmaceutical Technologies, National Institute of Chemical Pharmaceutical Research and Development—ICCF, 112 Vitan Av., 031299 Bucharest, Romania

**Keywords:** in silico study, DYRK2, neuroblastoma, phenylethanoids from olive oil, dietary lignans, verbascoside and pinoresinol, in vitro study, normal human astrocytes (NHAs) and human glioma cell line (U87), *Anemone nemorosa* ethanolic extracts

## Abstract

Neuroblastoma can be accessed with compounds of larger sizes and wider polarities, which do not usually cross the blood–brain barrier. Clinical data indicate cases of spontaneous regression of neuroblastoma, suggesting a reversible point in the course of cell brain tumorigenesis. Dual specificity tyrosine-phosphorylation-regulated kinase2 (DYRK2) is a major molecular target in tumorigenesis, while curcumin was revealed to be a strong inhibitor of DYRK2 (PBD ID: 5ZTN). Methods: in silico studies by CLC Drug Discovery Workbench (CLC) and Molegro Virtual Docker (MVD) Software on 20 vegetal compounds from the human diet tested on 5ZTN against the native ligand curcumin, in comparison with anemonin. In vitro studies were conducted on two ethanolic extracts from *Anemone nemorosa* tested on normal and tumor human brain cell lines NHA and U87, compared with four phenolic acids (caffeic, ferulic, gentisic, and para-aminobenzoic/PABA). Conclusions: in silico studies revealed five dietary compounds (verbascoside, lariciresinol, pinoresinol, medioresinol, matairesinol) acting as stronger inhibitors of 5ZTN compared to the native ligand curcumin. In vitro studies indicated that caffeic acid has certain anti-proliferative effects on U87 and small benefits on NHA viability. *A. nemorosa* extracts indicated potential benefits on NHA viability, and likely dangerous effects on U87.

## 1. Introduction

Among all cancers in humans, neuroblastoma (NB) impresses further by its predilection for an extremely fragile and emotional target, newborns, children and young people, respectively. Statistical data indicates that neuroblastic tumors comprise 8% to 10% of total childhood tumors, with 90% affecting children before the age of 5 [1,2]. According to data, NB has a genetic preset; the main genetic changes proved are the chromosome 1p36 deletion (around 70% of cases), chromosome 17q gain (around 80% of cases), 2p24*MYCN* proto-oncogene (myelocytomatosis viral oncogene neuroblastoma derived homolog) amplification (around 25% of cases), anaplastic lymphoma kinase oncogene (*ALK*) point mutations, or decreased expression of chromodomain-helicase-DNA-binding protein 5 (*CHD5)* level [2].

Considering the onset of the disease [3] and clinical data reporting regression cases of NB in the absence of any medical intervention [2,4], a preventive treatment can also be considered, especially in the context of the knowledge accumulated in the past years. The preventive treatment logically refers to pregnant women and nursing mothers. At the same time, the most feasible method of analysis consists of in silico studies on particular molecular targets resulting from clinical data. In this context, the most feasible NB inhibitors, with potential preventive activity, are specific plant compounds with antitumor potency found in every day food items, therefore, in the human diet; for example, dietary lignans (phenylpropanoid C6–C3 derivates) [5,6,7] and particular phenolic compounds from extra-virgin olive oil which have been proven to pass in the plasma of rats’ offspring [8]. The most feasible molecular targets are serine, threonine and tyrosine kinases, through which all phosphorylation-dephosphorylation processes in the eukaryotic cells are accomplished [9].

This way, clinical data appear to converge on the following feasible molecular targets: the tropomyosin kinase receptor homologs A, B and C (Trk) which control the spatiotemporal regulation of growth, differentiation, and survival of central and peripheral neurons [10]; the mammalian target of rapamycin (mTOR), a serine/threonine kinase which control cell growth, proliferation, survival and cell motility, and transcription, protein synthesis and autophagy [11,12]; 26S proteasome and its modulators (e.g., the tumor suppressor protein p53 and the dual-specificity tyrosine kinase2, DYRK2*),* which together play a critical role in tumor progression and tumor prevention by orchestrating a wide variety of cellular responses, including damaged cell apoptosis, maintenance of genomic stability, inhibition of angiogenesis, and regulation of cell metabolism and tumor microenvironment [13,14,15,16,17]; Aurora A serine/threonine kinase (AURKA), which also controls cell proliferation [18,19] is associated with poor prognosis in cancer tumors [20], specifically by increasing the expression of the vascular endothelial growth factor (VEGF) [21,22]; the mouse double minute 2 homolog (MDM2), which was proven to be a wide modulator of cell cycle and transcriptional events, including the activity of the tumor suppressor p53 [23,24], and DNA-topoisomerase I [25], the main treatment and molecular target in neuroblastoma.

The present paper presents docking studies on 20 natural vegetal compounds found in the human diet selected based on their unique ability to pass to offspring in rats and anti-tumor potency, studied on the dual-specificity tyrosine kinase DYRK2 molecular target (PDB ID: 5ZTN), in combination with its native ligand namely curcumin. The test compounds selected are oleuropein and ligstroside aglycones, oleacein, oleocanthal, tyrosol, elenolic acid, luteolin, apigenin, pinoresinol, lariciresinol, isolariciresinol, verbascoside, *o*-coumaric acid and vanillic acid from extra-virgin olive oil [8], and the main dietary lignans namely matairesinol, secoisolariciresinol, pinoresinol, lariciresinol, medioresinol and syringaresinol [5,6,7]. Studies were done in comparison with anemonin, another very active neuroprotective and antitumor [26,27,28,29] symmetrical plant molecule, also composed of two identical monomers. In silico studies are completed with in vitro studies, punctuated with an evaluation of the cytotoxic and anti-proliferative activity of two ethanolic extracts from fresh and dried *Anemone nemorosa* L. aerial part (*herba et flores*), on normal and tumor human brain cells, on normal human astrocytes NHA and human glioma cell line U87, respectively. Experimental studies are in the context of the impressive biological activity of anemonin in *Ranunculaceae* and *Anemone nemorosa* plant species [26,27,28,29], as well as of potential similitude with another toxic plant species, *Viscum album* L., very useful in cancer therapy today. In vitro studies were fulfilled in comparison with four vegetal test compounds, phenolic acids (reference compounds, Ref.) found in the human diet (caffeic acid, ferulic acid, gentisic acid and p-amino-benzoic acid/PABA), for a better understanding of plant compounds’ interaction with normal and tumor human brain cells and their potential benefits.

## 2. Results

### 2.1. In Silico Studies Results

In silico studies were done on 20 test vegetal compounds, dietary lignans and other phenolic compounds from extra-virgin olive oil proven to pass in offspring of rats and anemonin, respectively. The molecular target chosen for docking simulations was DYRK2 (PDB ID: 5ZTN) in complex with curcumin, its native ligand [30].

Appendix A show the complete data on the intermolecular interactions between the curcumin docked with 5ZTN in comparison with the 20 test vegetal compounds, by CLC Drug Discovery Workbench Software (QIAGEN Aarhus, Denmark) and by MVD Molegro Virtual Docker Software (Molexus IVS, Rørth, Denmark), using protocols previously established and validated [31,32].

Table 1 reviews the data in Appendix A. It presents the chemical structure of the 20 test compounds and the strength of the interaction (Docking Score and MolDock Score), ranked by the overall score they resulted from CLC and MVD analyses.

According to in silico study results (Table 1), by CLC software, verbascoside (score −88.14) is a stronger inhibitor of DYRK2 (PDB ID: 5ZTN) than the native ligand curcumin (score −83.92); by MVD software, verbascoside (score −148.78), lariciresinol (score −127.28), pinoresinol (score −126.85), medioresinol (score −124.68), and also matairesinol (score −116.98), are all stronger 5ZTN inhibitors in comparison with curcumin native ligand (score −115.85). Anemonin indicated the lowest interaction by CLC investigations (score −34.27) and the 13th position in the MVD study ranking (score −93.75), respectively. The common denominator of the most active vegetal test compounds in the study displays a pair of identical phenolic rings (interposed or not with a cyclopentane type ring) attached with several methanoic/ethanoic/propenoic acid chains providing a various number of active sites and –OH groups, necessary for the interaction with the aminoacids in the active pocket of the molecular target, DYRK2.

Appendix A, show the docking pose of the co-crystallized curcumin, the native ligand in the study, interacting with the amino acid residues in the binding site of 5ZTN, comparing the docking results of the 20 test vegetal compounds, by CLC and MVD software, respectively. Appendix A show the docking pose of the second reference compound in the study, anemonin, interacting with the amino acid residues in the binding site of 5ZTN by CLC and MVD study, too. It must be noted that anemonin does not form hydrogen bonds with the amino acids from the active site of the molecular target, 5ZTN, but with asparagine (ASP) 295. Its selection in the study was explained by its molecule symmetry (similar to curcumin), impressive biological activity, and ability to pass through the blood–brain barrier. 

Figure 1a,b shows the CLC docking pose of the most active DYRK2 inhibitor in this study, verbascoside ligand, respectively, its interactions with the amino acid residues found in the binding site of the molecular target (5ZTN), and the positions of the hydrogen bonds in the active pocket, particularly the interaction with glutamic acid (GLU) 193, lysine (LYS) 178, leucine (LEU) 231, serine (SER) 232, methionine (MET) 233, asparagine (ASN) 234 and LEU 231, SER 232, MET 233, ASN 280, ASN 234, LYS 178 and GLU 193 amino acids. Figure 2a,b shows the MVD docking pose of the interaction of verbascoside with the active aminoacids in the pocket, the hydrogen bond positions in the binding site of 5ZTN, particularly the interaction with GLU 193, LYS 178, ASN 234, ASN 280, ASP 285, ASN 285, GLU 279, isoleucine (ILE) and SER 232 amino acids. Figure 3 shows the steric interactions of the hydrogen bonds between the active hydroxyl groups in the verbascoside ligand and the active amino acids residues in the binding site of 5ZTN in 2D (MVD).

Table 2 shows the predicted drug-likeness parameters for the 20 test vegetal compounds under study in comparison to 5ZTN’s native ligand—curcumin.

The data series in Table 2 is the starting point for understanding the solubility of the test compounds, their transport behavior and their bioavailability in humans; therefore, the basic information for formulating decisions in the chemical-pharmaceutical industry. Among these, logP (the octanol-water coefficient) is the most important parameter since it provides indications of partition and lipophilicity of a compound in a medium [35]; it is used to predict the transport behavior and bioavailability of one particular compound in the digestive system, and subsequently tissues in humans. Accordingly, it was stated that “A drug targeting the central nervous system (CNS) should ideally have a logP value around 2. For oral and intestinal absorption, the ideal value is 1.35–1.8, while a drug intended for sub-lingual absorption should have a logP value > 5”. In addition, “A negative value for logP means the compound has a higher affinity for the aqueous phase (it is more hydrophilic); when logP = 0 the compound is equally partitioned between the lipid and aqueous phases; a positive value for logP denotes a higher concentration in the lipid phase (i.e., the compound is more lipophilic). LogP = 1 means there is a 10:1 partitioning in Organic: Aqueous phases” [36].

According to logP values in Table 2,with the exception of verbascoside, all 20 tested compounds exhibit a higher affinity for the lipid phase; *o*-coumarinic acid and isolariciresinol are feasible for CNS drug targeting; ligstroside (aglycone) and oleocanthal are feasible for oral and intestinal absorption; elenolic acid and anemonin will be equally partitioned between the lipid and aqueous phases; verbascoside prefers the aqueous phase, therefore, is the most feasible compound for bioavailability research studies in humans. Furthermore, lariciresinol, pinoresinol, medioresinol and matairesinol have zero violations of Lipinski’s criteria [37], another important parameter for the preparation of orally active drugs in the pharmaceutical industry. At the same time, verbascoside depicts three violations while simultaneously being the largest molecule in the study. Based on all this information, further in vivo studies and formulation decisions can be made.

### 2.2. In Vitro Pharmacological Studies Results

In vitro studies were conducted by Promega (Madison, WI, USA) MTS [3-(4,5-dimethyl -thiazol-2-yl)-5-(3-carboxymethoxyphenyl)-2-(4-sulfophenyl)-2H-tetrazolium]protocol [38], as cytotoxicity and anti-proliferative activity assays, respectively, and they were performed on four reference compounds (caffeic acid, ferulic acid, gentisic acid and para-aminobenzoic acid/PABA) series, and the two ethanolic extracts from *Anemone nemorosa* L. (AN1 and AN2) series, tested on normal human astrocytes and tumor human brain cells, NHA (CC-2565, Lonza, Basel, Switzerland) and U87 MG (ATCC-HBT-14, Tell City, IN, USA) cell lines, respectively. The four test reference compounds were prepared, each one as four dilution series (100, 50, 25 and 10 µg/mL sample, n = 4), and the two test vegetal extracts, AN1 and AN2, each one as seven dilution series (×1, ×2, ×5, ×10, ×50, ×100, and ×200, n = 7) in NHA and U87 culture medium; EMEM (Eagle’s minimum essential medium) with 10% heat-inactivated fetal bovine serum and 1% Penicillin–Streptomycin–Neomycin (PSN) antibiotic mixture for U87 MG cell line, and Astrocyte Growth Medium with Supplements, required for growth of the NHA line. In parallel, following an identical algorithm, four and, respectively, seven dilution series of the solvent sample, 70% ethanol, were prepared. The results were analyzed by comparison to the negative control series, represented by untreated control cells (n = 4, n = 7), and the positive control series, represented by 70% ethanol solvent series. Briefly, after reaching the cell confluence in the study (70% for cytotoxicity and 30% for anti-proliferative activity), the cells were detached from the flask with trypsin-EDTA and the resulting cell suspension was centrifuged at 2000 rpm for 5 min. Next, the cells were re-suspended in the individual growth medium; the cells were thenseeded in 96-well plates at a density of 4000 cells per well in 200 μL of the culture medium. Each test sample and corresponding positive control sample was applied to the cell cultures (triplicate series) in the dilution series described. After 20 h (in the case of cytotoxicity tests) and, respectively, 20 and 44 h (in the case of anti-proliferative tests) of cell exposure to the test and control samples, the culture medium was removed. The cells were then incubated with MTS solution for another two hours; after that, the viability of the adherent cells was determined by evaluating the absorbance of the solution at 490 nm. The recorded values (Optical Density, O.D. 490 nm) were used for cell viability estimation. It is also important to note that the ethanol solvent is chosen for its capacity to extract a large pallet of secondary metabolites from plants. In addition, it assures the complete solubilization of the test phenolic acids in the study as a basic condition for in vitro experiments. The comparison is all the more necessary as it has been demonstrated that the brain cells present sensitivity to ethanol; studies by mitochondrial neuronal health (MNH) assay [39] proved the toxicity of ethanol in the interval 9.2 µg–23 mg per sample, while the series in the present study are in the interval from 0.7 to 140 mg ethanol per sample.

#### 2.2.1. Cytotoxicity and Anti-Proliferative Activity of the Four Test Reference Compounds on Normal Human Astrocytes (NHA) and Human Glioma Cell Line (U87)

Figure 4A,B and Figure 5A,B show in vitro cytotoxicity and anti-proliferative activity results of the four test phenolic acids (caffeic, gentisic, ferulic and PABA), four dilution series each (100, 50, 25 and 10 µg/mL sample), on NHA and U87 cells. Overall, the comparison between the negative control test series (untreated cells) and positive control test series (cells treated with ethanol solvent in identical dilution series), yellow line versus red line, first of all, indicated NHA and U87 cell lines sensitivity to ethanol solvent in medium, in MTS cytotoxicity and anti-proliferative assay.

In this context, the cytotoxicity study on NHA after 24 h of cell exposure to the four test reference compounds, Figure 4A, overall indicated the capacity of caffeic acid, gentisic acid and PABA to inhibit the viability of the normal human brain cells in culture; in contrast, ferulic acid has revealed the ability to stimulate the proliferation of normal human astrocytes in vitro. In particular, the comparison of the test reference compounds series with the negative control series (yellow line) in the interval from 100 to 25 µg/mL sample indicated caffeic acid inhibitory effects of 77%, 50% and 34% (*p* > 0.05, n = 3), gentisic acid inhibitory effects of 71%, 65% and 51% (*p* < 0.05, n = 3) and PABA inhibitory effects of 75%, 50% and 41% (*p* < 0.05, n = 3).At the lowest concentration in the study, 10 µg/mL sample, caffeic acid and gentisic acid revealed small stimulatory effects estimated at 6% magnitude, while PABA indicated the ability to decrease the viability of the normal human astrocytes estimated at 4% inhibitory potency. On the other hand, ferulic acid stimulatory effects on NHA cells have been estimated at 7%, 21%, 25% and 30% (*p* < 0.05, n = 3). The comparison with the positive control series (red line) additionally highlighted the ability of ferulic acid to protect the normal human astrocytes in culture against the toxic action of ethanol in the environment throughout the concentration series studied.

The cytotoxicity study on U87 after 24 h of cell exposure, Figure 4B, revealed a different dynamic and cell response to the four test reference compounds. Overall, the lack of a stimulatory effect of ferulic acid on the test cells was observed, and the wide-ranging inhibitory effects of caffeic acid, gentisic acid and PABA on the proliferation of U87 in culture. The comparison of the test reference compounds series to the negative control series (yellow line) indicated caffeic acid inhibitory effects at the magnitude of 47%, 27%, 12% and 8% (*p* < 0.05, n = 3), gentisic acid at the magnitude of 31%, 21%, 11% and 7% (*p* < 0.05, n = 3), PABA at the magnitude of 45%, 8%, 3% and 3% (*p* > 0.05, n = 3), while ferulic acid inhibitory effects were estimated at 9%, 7%, 3% and 3% (*p* < 0.05, n = 3). Compared to the positive control series (red line), ferulic acid, but also caffeic acid, gentisic acid and PABA revealed potential cytoprotective activity against the highest concentration of ethanol in the environment.

The anti-proliferative MTS assay on U87, after 24 and 48 h of cell exposure to the four test reference compounds, Figure 5A,B revealed a similar dynamic and cell response observed in the cytotoxicity study on U87. Overall, the same lack of a stimulatory effect of ferulic acid on the human tumor brain cell in culture was observed in culture and a wide-ranging inhibitory activity of the four test reference compounds in the study. The comparison with the negative control point series (yellow line) after 48 h of cell exposure has shown the following results on the four test compounds series: caffeic acid inhibitory effects were estimated at the magnitude of 82%, 80%, 49% and 48% (*p* < 0.05, n = 3), gentisic acid at 40%, 33%, 11% and 9% (*p* < 0.05, n = 3), PABA at 70%, 57%, 16% and 9% (*p* < 0.05, n = 3), while ferulic acid inhibitory effects were estimated at the magnitude of 10%, 9%, 4% and 8% (*p* < 0.05, n = 3). Similarly, the comparison with the positive control series (red line) indicated all test compounds’ cytoprotective activity against the presence of ethanol in the environment. 

Furthermore, at the lowest concentration in the study, 10 µg/mL sample, caffeic acid was proven to have an anti-proliferative effect on U87 (calculated at 48% magnitude) and a cytoprotective effect on NHA (calculated at 6% magnitude); in this way, caffeic acid appears as a natural vegetal compound which has potential protective effects on the normal external human brain cells, and at the same time potential inhibitory activity on the external human tumor brain cells. Therefore, it could be considered a feasible human diet compound against neuroblastoma. At identical point series, ferulic acid has been emphasized with certain stimulatory activity on the viability of normal cells (estimated at 30% stimulatory activity on NHA) and less augmented anti-proliferative activity on tumor brain cells (estimated at 8% inhibitory activity on U87), at the same time with certain cytoprotective activity against the presence of ethanol in the environment.

#### 2.2.2. Cytotoxicity and Anti-Proliferative Activity of the Test Vegetal Extract AN1 on Normal Human Astrocytes (NHA) and Human Glioma Cell Line (U87)

Figure 6A,B and Figure 7A,B show the response of NHA and U87 after the treatment with crude ethanolic extract from fresh *A. nemorosa* (AN1), in MTS cytotoxicity and anti-proliferative assays.

The cytotoxicity MTS assay on NHA after 24 h of exposure to AN1, Figure 6A, and the comparison with the negative control series (yellow line) have revealed AN1 inhibitory effects on the cell viability at concentration levels higher than 17.6 µg gallic acid equivalents [GAE] per 1 mL test sample, versus AN1 stimulatory effects in the range between 17.6 and 0.44 µg [GAE]/mL test sample. The values of NHA cell viability in the study are presented as a percentage along the dilution series (*p* > 0.05, n = 3). These results could suggest potential toxic or potential beneficial effects of AN1, depending on the concentration level in the environment. The comparison with the positive control series (red line) also suggested potential cytoprotective activity against the toxic effects of ethanol solvent in the environment. The cytotoxicity study on the U87 cell line after 24 h of exposure to AN1, Figure 6B, overall confirmed the dynamic observed in cytotoxicity studies on NHA; inhibitory effects up to 17.6 µg [GAE]/mL test sample, followed by stimulatory effects down to 0.44 µg [GAE]/mL test sample, respectively. The values are also presented as a percentage along the dilution series (*p* < 0.05, n = 3). The comparison with the positive control series (red line) indicates the shift of the cytoprotective effect to lower concentrations of AN1 in the sample. 

The anti-proliferative MTS assay on U87 after 24 h and 48 h of exposure to AN1, Figure 7A,B, in comparison to the negative control series, together suggested the lability of the biological effects of AN1 on the tumor human brain cells in culture: MTS assay indicated that in the concentration range in which a clear inhibitory effect was determined at 24 h of exposure, a stimulatory effect at 48 h of exposure appeared, and vice-versa, which indicates the lack of control over the crude ethanolic extract from the fresh aerial part of *Anemone nemorosa*.

#### 2.2.3. Cytotoxicity and Anti-Proliferative Activity of the Test Vegetal Extract AN2 on Normal Human Astrocytes (NHA) and Human Glioma Cell Line (U87)

Figure 8A,B and Figure 9A,B show the response of NHA and U87 after the treatment with hot ethanolic extract from dried *A. nemorosa* (AN2) in cytotoxicity and anti-proliferative assays.

The cytotoxicity MTS assay on NHA after 24 h of exposure to AN2, Figure 8A, indicated similar inhibitory effects on the viability of normal brain cells in culture at concentration levels higher than 30.8 µg [GAE]/mL sample and stimulatory effects from 30.8 to 0.77 µg [GAE]/mL sample; the values are also presented as a percentage along the dilution series (*p* > 0.05, n = 3). The cytotoxicity study on U87 after 24 h of exposure to AN2, Figure 8B, overall indicated a more sustained capacity of AN2 to decrease the proliferation of the human tumor brain cells in culture, precisely in the interval from 154 to 3.08 µg [GAE]/mL sample; statistical calculation of the results on AN2 in comparison to the negative control series indicated statistically significant results on the U87 cell line (*p* < 0.05, n = 3). 

The anti-proliferative MTS assay on U87 after 24 h of exposure to AN2, Figure 9A, surprisingly confirmed the results noticed in the cytotoxicity study on U87, meaning the capacity of AN2 to statistically significantly sustain the decrease of U87 proliferation after 24 h of cell exposure (*p* < 0.05, n = 3). Further analysis of the anti-proliferative activity on U87 after 48 h of exposure to AN2, Figure 9B, unfortunately, led to the same conclusion that resulted in the AN1 study (the movement of the stimulatory and inhibitory effects of AN2 along the dilution series), confirming the lack of control over the hot ethanolic extract from dried *Anemone nemorosa* plant material as well, and ultimately the danger of using it to obtain both cytoprotective or antitumor effects.

Thus, by strictly following the conclusions from the cytotoxicity studies on NHA, the potential stimulatory effects on NHA of crude and hot ethanolic extract from fresh and dried *Anemone nemorosa* at concentration levels lower than 17.6 and 30.8 µg GAE/sample, respectively, these results could suggest AN1 and AN2 potential benefits on brain diseases associated with the decrease of the viability of neuronal cells.

### 2.3. Plant Extraction and Their Chemical Characterization 

The test plant material, *Anemone nemorosa* L. *herba et flores*, a common wood anemone, was collected in April 2021 from the Prahova region in the Romanian sub-Carpathians. Taxonomic identification has been conducted by the botanist’s team at the National Institute of Chemical-Pharmaceutical R&D, ICCF, Bucharest, Romania. A voucher specimen (dried plant) is available in the ICCF Plant Material Storing Room (code ANem21). 

The extraction process has been realized on fresh and dried aerial parts of the wood anemone as follows: (1) fifty (50) grams of fresh vegetal material was extracted with 1000 mL of cold 70% ethanol (*v*/*v*), 12 days at room temperature (16–18 °C), in dark conditions, and daily intermittent stirring; (2) ten (10) grams of dried vegetal material was extracted with 1000 mL of hot 70% ethanol, 60 min in a water bath at 80 °C, without stirring. Next, the vegetal mass was filtered on a double cotton filter. The resulting ethanolic extracts (codified AN1 and AN2) (Tracks T4 in Figure 10), but also the other three aqueous extracts (Tracks T2 in Figure 10) used in the author’s previous studies [40] were analyzed concerning chemical qualitative and quantitative aspects.

Qualitative analysis has been done by the high-performance thin-layer chromatography method (HPTLC), and Linomat 5 instrument (CAMAG, Muttenz, Switzerland), following the general method for polyphenols assessment in plant-derived products [41]: solvent system (mobile phase)-ethyl acetate:acetic acid:formic acid:water, 100:12:12:26; identification-immersion in 1% diphenylboric acid β-aminoethyl ester complex in methanol, followed by 5% polyethylene glycol 4000 in ethanol (Natural products reagent, NP/PEG No. 28) and exposure to UV-366 nm. 

Figure 10 reveals the major polyphenol compounds in aqueous (T2 Tracks) and ethanolic (T4 Tracks) extracts from fresh and dried aerial parts of *Anemone nemorosa* L. (*herba et flores*), in comparison with several notorious plant phenolics, reference compounds (ref.), respectively; T4.1 corresponds to the crude ethanolic extract from the fresh plant (AN1), and T4.2 corresponds to the hot ethanolic extract from the dried plant (AN2).

The fingerprint analysis, coloration and position of the spots in Anemone *nemorosa* polar extracts depicted in Figure 10 indicate the high similitude of the aqueous and ethanolic extracts from fresh and dried A. *nemorosa* plant material, at the same time, the impressive number of polyphenols compounds in wood anemone polar extracts. Specifically, the major polyphenol compounds in wood anemone aqueous and ethanolic extracts are caffeic acid derivates (blue fluorescent/fl. spots s1, s4, s6, s7, s8, s9, s10), most of them found in esterified form in spring time, but also augmented quantities of quercetin derivates (yellow-orange fl. spots s3, s4) and apigenin derivates (dark green, fl. spot s2 and s11); the blue fl. spot at the START position (s0) indicates ellagitannin presence, while the indigo fl. spot on the top of the chromatogram (s12) likely indicates the presence of protocatechuic acid. 

Quantitative analysis used the Folin–Ciocalteau method to estimate the total phenols content in samples [40]; the results were rendered as mg total phenols content expressed as gallic acid equivalents [GAE]per 1 mL sample. Accordingly, AN1 was estimated with a total phenol content of 88 mg GAE/mL extract, while AN2 showed 154 mg GAE/mL extract. More detailed compositional data on *Anemone nemorosa* L. plant species are available in recent years’ comprehensive literature [29].

## 3. Discussion

According to the literature data, the 26S proteasome is a nodal point in cell eukaryote division, function and apoptosis processes [42,43]; therefore, it is highly interested in cancer therapy and cancer research. In addition, 26S proteasome is described as a *molecular machine* composed of about 31 protein subunits which assure the degradation of the (defective) proteins resulting from normal cell processes such as division, differentiation, migration and apoptosis, and pathological events such as inflammation or skeletal muscle proteolysis, but also from modified transcription processes occurring in tumorigenesis. The core of the 26S proteasome, namely the 20S proteasome, is a barrel-shaped proteolytic complex capped by two units with regulatory activity, namely the 19S proteasome. The 19S proteasome cap units actually recognize and bind to the defective marked proteins (with poly-ubiquitin chains); after that, they are conducted inside the hydrolytic area of the 20S proteasome. The degradation process in the 20S proteasome is coupled with caspase activity, and according to available data [44], this process can occur via the ubiquitin-dependent pathway using 19S regulatory units but also by simply recognizing the N-and C-terminal caspase-cleaved peptides and directly tracking them into the hydrolytic area of the 20S proteasome. The data also points out that if caspase cleavage sites are peptide segments rich in proline (P), glutamic acid (D), aspartic acid (E) and serine (S) or threonine (T) residues, namely PEST motifs, the 26S/20S proteasomes’ proteolytic cleavage sites are similar to those of trypsin and chymotrypsin in the digestive system.

Therefore, due to its major role in signaling pathways of normal and pathological cell cycles and cell death, the 26S proteasome is a primary molecular target in cancer therapy. Bortezomib (I) is the first antitumor compound designed to inhibit 26S proteasome [45,46]; it is a dipeptidyl boronic acid and a member of pyrazines and L-phenylalanine derivatives able to interrupt cell division, to induce cell apoptosis and to inhibit the nuclear factor-kappa B (NF-κB). It is principally used in multiple myeloma but also breast cancer and neuroblastoma. The main disadvantage and weakness of Bortezomib is the fact that it induces tumor resistance to chemical treatment. 

Relative to neuroblastic tumors, data [47] indicate its capacity to induce an extraordinary proliferative potential of the cells; at the same time, methylation changes in the tumor cells are considered the main causes of resistance to Bortezomib. 



For the same purpose, the downstream and upstream inhibitor or activator molecules of 26S/the 20S/19S proteasomes are of high interest in the cancer approach. This way, the DYRK2, a major adenosine triphosphate (ATP)-dependent phosphorylase in eukaryotic cells, has also been proven as a feasible upstream molecular target in cancer therapy. Moreover, DYRK2 acts as a strong activator of 26Sproteasome. Briefly, DYRK2 is a priming kinase in the posttranslational signaling and regulatory pathway in cells, acting through reversible catalyzes of hydroxyl-phosphorylation of tyrosine/Tyr, serine/Ser, or threonine/Thr residues of numerous protein substrates. The major substrates of DYRK2 are 26S proteasome, tumor suppressor protein p-53, trans-membrane neurogenic locus notch homolog protein 1 (NOTCH1), heat-shock factor 1 protein (HFS), as well as c-Myc and c-Jun (myelocytomatosis and AP-1 transcription factor subunit) protooncogenes, proteasome component Rpt3, telomerase reverse transcriptase (TERT), and the catalytic subunit of a complex which severs microtubules in cells namely katanin p60, each one revealed as deregulated in cancer tumors [48,49]. Altogether, DYRK2 is assumed with a putative role in cells since it can be both pro-tumorigenic signaling by stimulating cell growth and antitumorigenic signaling by stimulating cell apoptosis.

Among potential DYRK2 inhibitors, studies nowadays converge on harmine and curcumin [49] plant compounds [50,51].

Harmine (II), a carboline alkaloid derivative from *Peganum harmala*, is mainly known for its anti-platelet, anti-thrombotic and vasodilatory activities [52] through inhibiting cyclic adenosine monophosphate (AMP) phosphodiesterase and protein tyrosine phosphorylation in human vessel tissue. In addition, harmine alkaloid has been shown to have anti-proliferative activity upon myeloid leukemia [53]. 

Considering the key role of the methylation process in the induction of neuroblastoma resistance to Bortezomib, it is of high interest given the fact that the in vitro studies have proved harmine inhibitory potency, precisely through the modulation of DNA methyltransferase in acute promyelocytic leukemia NB4 cells; the effects on NB4 were in a time and dose-dependent manner, with doses of 102 µg harmine alkaloid per 1 mL sample significantly increasing the amount of the cells in G1 Phase [53]. 



Curcumin (III), another notorious plant compound from *Curcuma longa*, has also been proven to have a plethora of human health benefits [54], while in vitro [54,55] and in silico [30] studies strongly sustain curcumin antitumor potency and anti-neuroblastoma activity; yet, very low solubility and cellular uptake make its use very difficult. Thus, the current research is directed at nanoformulation drug-type vehicles, proven to have better bioavailability in humans. Among the multiple tumor and cell types studied, in vitro studies on the neuroblastic lines indicated curcumin’s ability to restrain serine-threonine kinase (Akt) and NF-κB activities, to promote mitochondrial dysfunction, and to upregulate the tumor suppressor protein p53 and caspase signaling too [55]. Moreover, some recent in silico studies indicated DYRK2 prime kinase phosphorylase as a direct target of curcumin [30]. 

The interaction between DYRK2 and the tumor suppressor p53 phosphorylation substrate has also been revealed to be of high interest in managing human brain tumors; the main argument is that p53 decreased in 84% of glioblastoma patients and 94% of total tumor brain cell lines [14]. 

According to the literature, DYRK2 stimulation induces phosphorylation at Serine 46 (Ser46) in tumor suppressor p53; the activated p53 induces antitumor effects by multiple genes and proteins activation by the cell-cycle arrest in G1, and by increasing tumor cell apoptosis too [48]. Moreover, the point mutations or deletions at the tumor protein gene *TPp53* were associated with poor survival prognosis in brain tumors. A stoichiometric relationship between p53 activation and phosphorylation at Ser46 has been revealed; hence, the *TPp53* mutations were framed as pro-oncogenic in nature [14,48]. The presence of the gain-of-function (GOF) mutant p53 gene, *GOFp53,* was associated with decreased phosphorylation at Ser46; therefore, *GOFp53* mutant variant was declared the major promoter of brain tumors in humans. The mutant variant is associated with an improper, wrong folding of p53 resulting in unstable complexes and weak cell signals to the pro-apoptotic targets downstream [14]. 

Furthermore, the activity of the tumor suppressor p53 in normal cells is controlled by binding the mouse double minute 2 (MDM2) homologs [17]. This fact was proved with certainty by in vitro experiments in the presence of compound Nutlin-3 (IV), an activator of p53; Nutlin-3 induced a dose-dependent increase of some of the most important indicators of apoptosis in cells (the p53 upregulated modulator of apoptosis PUMA, poly-polymerase 1 PARP and protein p21 which react first to DNA damage) at the same time it stopped cell growth in S phase [15,17]. 

The interaction between tumor suppressor p53 and MDM2 occurs by binding the 106–amino acid-long N terminal domain of MDM2 with the N terminal transactivation domain of p53 [56,57]. This way, two feasible therapies and pathways in managing brain tumors in humans have been established: either by stabilizing p53 conformation as indicated in studies using PRIMA-1 (V), ((2,2-bis(hydroxymethyl)-1-azabicyclo [2.2.2]octan-3-one)) [14], or by inhibiting, or disrupting p53 interaction with MDM2 ligand, as indicated in the experiments using Nutlin-3, MI-219 (VI) [17], but also various vegetal ceramides [16], curcumin [30,55], chalcones [17,58,59], and other compounds found in the specialized catalogs [60].



Above all, it is well known that the rapid growth of the tumor mass causes a lack of vascularization inside the tumor. Therefore, hypoxia and necrosis processes frequently occur inside the tumor mass. Haupt et al. [61] and Ghahremani et al. [62] have made an important contribution to understanding the interrelation between tumor suppressor protein p53 and vascular epidermal growth factor (VEGF).

According to their findings [61,62], the cancer tumor can adapt to hypoxia by mutations that promote neo-angiogenesis or by promoting a new balance between p53 and VEGF. Furthermore, the tumor suppressor p53 has been revealed with a double role in the relationship with VEGF; by binding to the existing site in the VEGF, p53 accelerated the transcription of VEGF in hypoxia, but if the hypoxia was prolonged, p53 reduced VEGF expression promoting cell apoptosis, following the retinoblastoma pathway in a p21-dependent manner. By data results corroboration, the authors [60,61], in fact, demonstrated the danger coming from the dual role (stimulator and inhibitor) of p53 and its mutant (proved present in 50% of all types of cancer in humans) on VEGF. Furthermore, their results clearly demonstrated that if p53 stimulation is made in the context of increasing levels of VEGF (resulting in normal tumor progression), the patient’s condition worsens, while p53 stimulation in the context of using a topoisomerase I inhibitor results in increasing effectiveness of the cancer treatment—hence the major interest, but also the major concern in p53 modulation to control tumor progression.

Finally, the high efficacy in neuroblastoma of DNA-topoisomerase I inhibitors (e.g., camptotechin and its derivatives irinotecan and topotecan) was demonstrated decades ago [63,64,65]; at the same time, there were many chemical–chemical and chemical–natural combinations tested to manage tumor resistance or to obtain synergistic activity [66,67,68]. In addition, in silico studies have also revealed new molecular targets and tumor inhibitors as potential cancer approaches and combined therapies [69,70,71].

Considering the major role of 26S proteasome in tumorigenesis and the complexity of the molecular machine 26/20/19S proteasomes, meaning three active inhibitors to suppress its activity, at the same time, the potential danger from tumor suppressor p53 modulation, due to potential VEGF stimulation—DYRK2 inhibition appears as the best option in studying new antineoplastic agents, more so as its native ligand in PDB Bank is the notorious plant compound curcumin [30]. Therefore, comparing the 20 active compounds in the human diet results predicts potential preventive anti-neuroblastoma agents.

Furthermore, the fact that 14 of the 20 test vegetal compounds (e.g., oleuropein and ligstroside aglycones, oleacein, oleocanthal, tyrosol, elenolic acid, luteolin, apigenin, pinoresinol, lariciresinol, isolariciresinol, verbascoside, *o*-coumaric acid and vanillic acid from extra-virgin olive oil) were shown to pass into the plasma of rats’ offsprings is another argument in support of the opportunity of in silico studies. In addition, the comparison with anemonin answers another specific criterion of brain tumors, the ability to pass through the blood–brain barrier.

In this context, in silico studies indicated that DYRK2 common denominator inhibitors have a (poly)(hydroxyl)phenyl-methanoic/ethanoic/propenoic acid ring, providing the active sites and –OH groups necessary for the interaction with the active amino acids in the pocket of the molecular target DYRK2 (PDB ID: 5ZTN).

Among the 20 test vegetal compounds, CLC and DVM software’s docking scores in comparison to the native ligand curcumin, both indicated the verbascoside compound as being a stronger inhibitor of DYRK2 (5ZTN); the results are −88.14 versus −83.92, and −148.78 versus −115.85, respectively. It is well-known that verbascoside is a water-soluble phenylethanoid glycoside first isolated from mullein (*Verbascum thapsus* L.). Still, it can also be found in many other plant species, as well as in the mill wastewater polluting byproduct from the olive oil production process [72].

According to data reported, verbascoside has augmented scavenger, antioxidant, and anti-inflammatory properties and chemo-preventive, wound-healing, and neuro-protective effects [72,73]. Furthermore, even if it is unable to cross the blood–brain barrier in humans (see Table 2, due to a negative logP value and large molecular mass), neuroblastoma has the advantage of being accessed by compounds of larger sizes and wider polarities. Therefore, verbascoside appears as a feasible anti-neuroblastic agent, at the same time, a potential preventive treatment.

In DVM software docking scores, lariciresinol (score −127.28), pinoresinol (score −126.85), medioresinol (score −124.68), and matairesinol (score −116.98) also were proved as stronger inhibitors of 5ZTN in comparison to the native ligand curcumin (score −115.85). They all are dietary lignans, while lariciresinol and pinoresinol are also found in extra-virgin olive oil, with a very high rate of interest for human health.

In terms of bioavailability in humans, the results in Table 2 suggest their affinity for the lipid phase, which can be speculated as easily passing through the cell wall. In fact, the in vitro studies on secoisolariciresinol diglucoside, secoisolariciresinol, pinoresinol, lariciresinol, matairesinol, and hydroxymatairesinol (reference compounds) tested on human colon cancer cell line Caco-2 have revealed all these lignans as being absorbed and metabolized by intestinal cells. Among these, pinoresinol was the most efficiently conjugated and taken up by the intestinal cell (75% of the total in cells); at the same time, it has exhibited the strongest anti-inflammatory activity by acting on the NF-κB signaling pathway [74]. Accordingly, pinoresinol, another plant compound found in the human diet, was also passed to the rats’ offspring, which appears as a feasible DYRK2 inhibitor with potential preventive activity against neuroblastoma.

Figure 11 presents a comparative, general view of the docking and MolDock scores of the 20 test vegetal compounds in the study.

Anemonin from the *Anemone nemorosa* plant species and other species of the *Ranunculaceae* family showed the lowest position in the CLC ranking and 13th position after curcumin in MVD, indicating a decreased potential as preventive or direct anti-neuroblastic activity.

In vitro MTS cytotoxicity and anti-proliferative assays on normal human astrocytes (NHA) and a human glioma cell line (U87) after 24 h, and 24 h and 48 h, respectively, of exposure to the four test reference compounds (caffeic acid, gentisic acid, ferulic acid and para-aminobenzoic acid/PABA) together concluded caffeic acid’s anti-proliferative potential on U87 (e.g., 48% inhibitory activity on the cell viability), at the same time small benefits on NHA (e.g., 6% stimulatory activity on the cell viability). These results were calculated at 10 µg test compound per sample, therefore, achievable from the human diet. Furthermore, at identical test concentrations, ferulic acid has been emphasized with stimulatory activity on the viability of normal human brain cells in culture (e.g., 30% stimulatory activity on NHA) and less augmented anti-proliferative effects on human tumor brain cells in culture (e.g., 8% inhibitory activity on U87), at the same time with certain cytoprotective activity against the presence of the ethanol solvent in the environment.

In vitro cytotoxicity and anti-proliferative MTS assays on crude and hot ethanolic extracts from fresh and dried *Anemone nemorosa* aerial parts (*herba et flores*), AN1 and AN2, tested on NHA have revealed potential stimulatory versus potential inhibitory activity function of their concentration in samples. MTS anti-proliferative tests on U87 cells after 24 and 48 h of exposure to AN1 and AN2 have revealed that in the concentration range in which a clear inhibitory effect was determined after 24 h of exposure, a stimulatory effect at 48 h of exposure appeared, and vice-versa, therefore, suggesting the lability of the biological effects on U87 and potential danger in using them against neuroblastoma.

## 4. Materials and Methods

### 4.1. In Silico Assay

In silico, molecular docking study was conducted to predict the probable binding site and mode of binding of the 21 test vegetal compounds with dual-specificity tyrosine-regulated kinase 2, DYRK2, (PDB ID: 5ZTN) [30]. The docking studies have been performed using CLC Drug Discovery Workbench Software (QIAGEN, Aarhus, Denmark) and MVD Molegro Virtual Docker Software (Molexus IVS, Rørth, Denmark) protocols, as previously described [31,32]. As a general concept, molecular docking allows an accurate prediction of the optimized conformation of tested compounds (as ligands) and their target receptor protein to achieve a stable complex.

The score and hydrogen bonds formed with the amino acid residues from the binding site are further used to predict the binding modes, the binding affinities, and the orientation of the docked ligands in the active site of the protein receptor, DYRK2, respectively. In this work, the structures of the 21 compounds were imported from PubChem database [33] and prepared by energy minimization with Spartan’18 software [75] using molecular mechanics force field (MMFF) [34]. The co-crystallized curcumin and anemonin were taken as reference compounds to compare the docking results of the other test compounds. The calculations have been performed on the optimized structures of molecules, thus presenting the configuration of minimum energy and, accordingly, an optimized geometry in vacuum conditions without any solvent corrections. Briefly, the docking protocol involved the following steps: ligands and protein preparation, removal of co-factors and water molecules; setup of binding site and binding pocket; docking simulations on co-crystallized and investigated ligands; validation, collecting property data and docking results. Interactions by hydrogen bonds of ligands within the active binding site of the studied protein target were identified and measured. The results were given in terms of docking scores by the used software, CLC Drug Discovery Workbench or Molegro Virtual Docker, respectively.

Additionally, a drug-likeness analysis was done based on structural parameters correlated with Lipinski’s rule of five [76]. The results of the 21 test compounds in the study were compared and discussed.

### 4.2. In Vitro Pharmacological Studies

In vitro studies were performed by MTS assay following CellTiter 96AQueous One Solution Cell Proliferation Assay Promega Corporation (Madison, WI, USA) protocol [38]; as a general principle, this is a colorimetric test based on the selective ability of the viable cells in culture to reduce the tetrazolium component of MTS [3-(4,5-dimethylthiazol-2-yl)-5-(3-carboxymethoxyphenyl)-2-(4-sulfophenyl)-2H-tetrazolium] in medium into a purple coloured formazan crystal which can be further measured at 490 nm. Depending on how the MTS assay is conducted, it is possible to evaluate the cytotoxic activity or the anti-proliferative activity of the test samples in comparison with negative control and positive control series samples. The difference between the two types of MTS assays is that in the cytotoxicity study, the cells are exposed to the test and control samples at the time when “semiconfluent” cell culture is achieved (meaning about 70% cell proliferation), while in the anti-proliferative study, the application of the test and control samples is conducted at the time when “sub-confluent” cell culture is achieved (meaning about 30% cell proliferation). Broadly, in vitro assay aims to investigate the cytotoxicity and anti-proliferative potential on human normal and human tumor brain cells of three series of test samples: test reference compounds series comprising four phenolic acids found in human diet, caffeic acid, ferulic acid, gentisic acid and para-aminobenzoic acid/PABA, respectively. The crude ethanolic extract from fresh *Anemone nemorosa* plant material (*herba et flores*) codified AN1 series, and the hot ethanolic extract from dried *Anemone nemorosa* plant material (*herba et flores*) codified AN2 series. Notably, the four test reference compounds series were prepared as four dilution series (100, 50, 25 and 10 µg/mL sample) in culture medium, starting from a stock solution of 1 mg/mL in 70% ethanol (*v*/*v*). For the two test vegetal extracts, AN1 and AN2 series, each one was prepared as seven dilution series (×1, ×2, ×5, ×10, ×50, ×100, ×200) in culture medium, starting from the initial 70% crude and hot ethanolic extract with a content of 88 µg GAE/mL in the case of AN1, and 154 µg GAE/mL in the case of AN2. In parallel, following an identical algorithm, four and, respectively, seven dilution series of the solvent sample, 70% ethanol, have been prepared as positive control series. Accordingly, MTS tests were performed on human normal brain cell line NHA (CC-2565, Lonza, Basel, Switzerland) and human tumor brain cell line U87 MG (ATCC-HTB-14, Tell City, IN, USA). Briefly, after reaching the particular cell confluence in the study (70% for cytotoxicity and 30% for anti-proliferative activity), the cells were detached from the flask with trypsin-EDTA. The cell suspension was further centrifuged at 2000 rpm for 5 min after that, re-suspended in the growth medium: EMEM (Eagle’s minimum essential medium) with 10% heat-inactivated fetal bovine serum and 1% Penicillin–Streptomycin–Neomycin (PSN) antibiotic mixture, for U87 MG cell line and Astrocyte Growth Medium with Supplements, required for growth of astrocytes; after that, the cells were seeded in 96-well plates at a density of 4000 cells per well in 200 μL of the culture medium. Each test sample and corresponding positive control sample in the dilution series were applied to the cell cultures in triplicate series. After 20 h (in the case of cytotoxicity test) and, respectively, 20 and 44 h (in the case of anti-proliferative test) of cell exposure to test and control samples, the culture medium was removed. The cells were then incubated with MTS solution for another two hours; after that, the viability of the adherent cells was determined by evaluating the absorbance of solution at 490 nm (Chameleon V Plate Reader, LKB Instruments). The results, Optical Density (O.D. at 490 nm) values and the viability percentages along the series (n = 3) were computed for their statistical significance (Student *t-*test); note: *p* > 0.05 means results without statistical significance; *p* < 0.05 means results with statistical significance.

The cell culture reagents EMEM (Eagle’s minimum essential medium) and Penicillin–Streptomycin–Neomycin (PSN) Antibiotic Mixture were purchased from Thermo Fisher (Waltham, MA, USA) Distributor in Romania), and Astrocyte Growth Medium with Supplements and the Fetal Bovine Serum (FBS) were purchased from Merck (Sigma-Aldrich, Saint Louis, MO, USA) Distributor in Romania.

### 4.3. Plant Extract Preparation

The plant material, *Anemone nemorosa* L. *herba et flores*, was collected in April 2021 from the sub-Carpathian region Prahova in Romania. Taxonomic identification was assured by the botanist’s team at the National Institute for Chemical-Pharmaceutical R&D, ICCF, Bucharest, Romania. A voucher specimen (codified ANem21), dried plant, is deposited in ICCF Plant Material Storing Room. Briefly, fifty (50) grams of fresh plant material was extracted with 1000 mL of 70% ethanol (*v*/*v*), 12 days at room temperature (16–18 °C), with intermittent (daily) stirring; another ten (10) grams of medium size dried plant material was extracted with 1000 mL of 70% ethanol (*v*/*v*), 60 min at the reflux temperature (80 °C). After the extraction time, the plant mass extract was passed through a double cotton filter. Two ethanolic extracts were obtained: a crude/cold extract from fresh *Anemone nemorosa* plant material *(herba et flores*) codified AN1 and a hot ethanolic extract from dried *Anemone nemorosa* plant material (*herba et flores*) codified AN2.

The extracts, AN1 and AN2, were disposed of in two brown glass bottles and used in further chemical analytical characterization and in vitro pharmacological tests.

### 4.4. Plant Extracts Chemical Characterization

The two test vegetal extracts, AN1 and AN2, were characterized as chemical qualitative and quantitative.

#### 4.4.1. Chemical Qualitative Characterization of Plant Extracts

Chemical qualitative characterization of the test vegetal extracts was accomplished by high-performance thin-layer chromatography (HPTLC) and Linomat 5 instrument (CAMAG, Muttenz, Switzerland). It was used as the general method for polyphenols assessment in vegetal samples [41]; solvent system-ethyl acetate: acetic acid:formic acid:water, 100:12:12:26, respectively. Briefly, test vegetal extracts in volumes from 1 to 5 μL were automatically applied as 8 mm band length in the 10 × 10 cm Silica gel 60F HPTLC plate (Merck, Darmstadt, Germany) using a Hamilton syringe; test reference compounds (ref., prepared as 10^−3^ M solutions in 70% ethanol, *v*/*v*) and their mixtures were also applied in 0.5–2.5 μL per 8 mm band. The loaded plate was then kept in TLC twin developing chamber at 16–18 °C with the particular solvent system (mobile phase) for polyphenols assessment, up to 90 mm. The developed plate was carefully dried with a hair dryer and then immersed into the NP/PEG No. 28 identification reagents. The plate was next disposed of inside the Photo-documentation chamber of CAMAG Linomat 5 apparatus, and the image at UV-366 nm was captured. After specific chromatogram Rf measurements, polyphenols spots were assigned by comparison to literature data and particular test reference compounds used in the study.

#### 4.4.2. Chemical Quantitative Characterization of Plant Extracts

Chemical quantitative characterization of the test vegetal extracts intended total phenolics content assessment in samples. It was used as the standard *Folin–Ciocalteau* method, as described in the previous authors’ studies on *Anemone nemorosa* polar extracts [40], and UV/Vis Hélios γ (Thermo Electron Corporation, Waltham, MA, USA) spectrophotometer. Briefly, (three) aliquots of 10–500 μL test vegetal extract are mixed with 200 μL of *Folin–Ciocalteau* reagent and accurately finished at 5000 μL volumetric flasks with (5%, *w*/*v*) sodium carbonate. Flasks were mixed and left in a dark place at room temperature for 30 min, and then the absorbance at 750nm was measured. The content of total phenols in samples is estimated by comparison with gallic acid (reference substance) calibration curve: R*^2^* = 0.998, n = 3. Results are therefore expressed as mg gallic acid equivalents [GAE] per 1 mL test sample.

#### 4.4.3. Chemicals, Reagents and Reference Compounds Description

The chemicals (sodium carbonate), reagents (*Folin–Ciocalteau*, *Natural Product*, *PEG*), solvents (ethanol, ethyl acetate, formic acid, and glacial acetic acid), and reference compounds (e.g., quercetin-3-O-rutinoside/rutin (min. 95%), quercetin-3-O-galactoside/ hyperoside (>97%), apigenin-7-O-glucoside/cosmosiin (>97%), apigenin-8-C-glucoside/ vitexin (>96%), apigenin (97%), kaempferol (95%), caffeic acid (99%), chlorogenic acid (>95%), gallic acid (95%), protocatechuic acid (97%), rosmarinic acid (>99%)) used in the study were purchased from Sigma-Aldrich (Saint Louis, MI, USA) Distributor in Romania.

## 5. Conclusions

By CLC and MVD in silico studies, three components of extra-virgin olive oil, verbascoside, lariciresinol, and pinoresinol, and two dietary lignans, medioresinol and matairesinol were proved as stronger DYRK2 (PDB ID: 5ZTN) inhibitors than the native ligand curcumin. Their common chemical denominator consists of a pair of identical phenolic rings (interposed or not with a cyclopentane type ring) attached to several methanoic/ethanoic/propenoic acid chains providing the active –OH groups for devising hydrogen bonds with the active amino acids in the binding site of the molecular target 5ZTN. According to the MVD study, the hydrogen bonds between the most active compound in the study, verbascoside, and the amino acids in the 5ZTN binding site are O sp^3^ (O15)-N sp^3^ from LYS 178, O sp^3^ (O15)-O sp^3^ from GLU 193, O sp^3^(O1)-N sp^2^ from ASN 234, O sp^3^ (O9)-O sp^2^ from ASN 280, O sp^3^ (O9)-O sp^3^ from ASP 295, O sp^3^ (O8)-O sp^2^ from ASN 280, O sp^3^ (O8)-O sp^2^ from GLU 279, O sp^3^ (O6)-O sp^2^ from ILE 155, O sp^3^ (O12)-O sp^3^ from SER 232 and O sp^3^ (O13)-O sp^3^ from SER 232; in CLC study, they are O sp^3^ (O8)-O sp^2^ from LEU 231, O sp^3^ (O9)-O sp^3^ from SER 232, O sp^3^ (O9)-O sp^2^ from MET 233, O sp^2^ (O11)-N sp^2^ from ASN 234, O sp^2^ (O11)-N sp^2^ from ASN 234, O sp^3^ (O13)-N sp^3^ from LYS 178 and O sp^3^ (O13)-O sp^2^ from GLU 193.

In vitro MTS assays on normal human astrocytes (NHA) and a human glioma cell line (U87) after exposure to the four test phenolic acids (caffeic, gentisic, ferulic and para-aminobenzoic acid/PABA) commonly found in the human diet, together concluded caffeic acid’s anti-proliferative potential on U87 (estimated at 48% inhibitory activity), at the same time cytoprotective activity on NHA (estimated at 6% stimulatory activity), at the lowest dilution series in the study (10 µg/mL sample), therefore achievable from the human diet (mainly from cocoa-derived products). On the other hand, ferulic acid (at identical 10 µg/mL sample) has revealed certain stimulatory activity on NHA (estimated at 30% potency) and less augmented anti-proliferative activity on U87 (estimated at 8% potency), at the same time with certain cytoprotective activity against the presence of ethanol in the environment.

In vitro MTS assays on NHA and U87 after the exposure to AN1 and AN2 crude and hot ethanolic extracts from fresh and dried *Anemone nemorosa*, both experiments have revealed potential stimulatory versus potential inhibitory activity of AN1 and AN2 function on their concentration in samples. Together, the cytotoxicity assays on NHA concluded potential cell viability and stimulatory effects at concentration levels lower than 17.6 and 30.8 µg GAE/mL, respectively. This suggests potential benefits for brain diseases associated with the decrease of the viability of neuronal cells. On the other hand, the anti-proliferative assays on U87 have revealed the lability of the stimulatory and inhibitory effects of AN1 and AN2 along the dilution series; therefore, the lack of control over the effects of crude and hot ethanolic extracts from fresh or dried *Anemone nemorosa*, and no security in using them against neuroblastoma.

In conclusion, while in silico studies indicated five vegetal compounds from the human diet, verbascoside, lariciresinol, pinoresinol, medioresinol, and matairesinol, with potential activity against DYRK2 and neuroblastoma, in vitro MTS assays suggested caffeic acid’s potential inhibitory activity against the proliferation of tumor brain cells, and at the same time potential stimulatory activity upon normal astrocytes in humans.

## Figures and Tables

**Figure 1 ijms-24-07404-f001:**
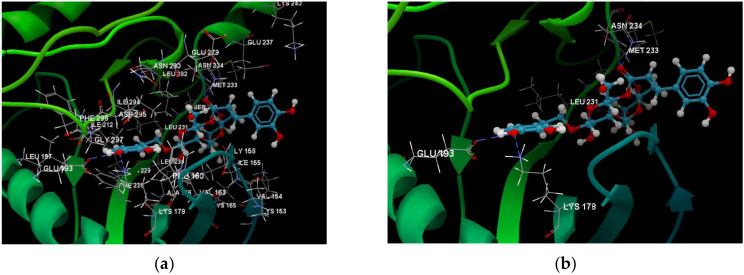
Interactions between verbascoside and the amino acid residues from the active binding site of 5ZTN: (**a**) general docking image of the verbascoside ligand interacting with the amino acid residues of the binding site of 5ZTN; (**b**) the hydrogen bonds between verbascoside and GLU 193, LYS 178, LEU 231, SER 232, MET 233, ASN 234 and ASN 280 active amino acids (CLC).

**Figure 2 ijms-24-07404-f002:**
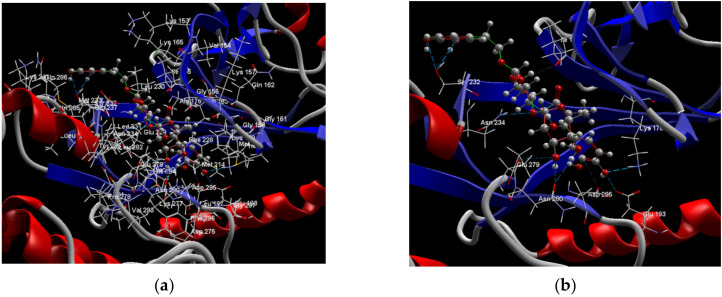
Interactions between verbascoside and the amino acid residues from the active binding site of 5ZTN: (**a**) a general docking image of the verbascoside interacting with the amino acid residues of the binding site, and (**b**) the hydrogen bonds between verbascoside and LYS 178, GLU 193, ASN 234, ASN 280, ASP 295, GLU 279, ILE 155, and SER 232 active amino acids (MVD).

**Figure 3 ijms-24-07404-f003:**
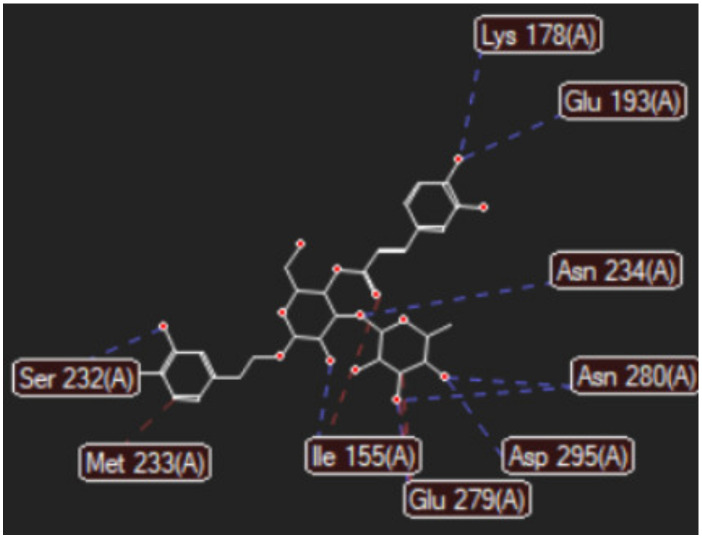
The hydrogen bonds (blue) and steric interactions (red) between the verbascoside ligand and the active amino acids residues of the binding site of 5ZTN in 2D (MVD).

**Figure 4 ijms-24-07404-f004:**
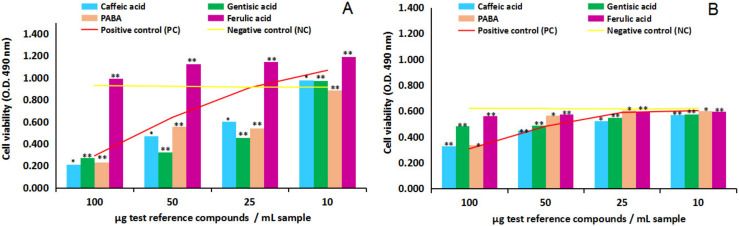
Effect of test reference compounds (caffeic acid, gentisic acid, ferulic acid and para-aminobenzoic acid/PABA) on human normal brain cell line NHA and human tumor brain cell line U87, in cytotoxicity MTS assay. (**A**) NHA cell line viability after 24 h of exposure to test reference compounds. (**B**) U87 cell line viability after 24 h of exposure to test reference compounds. Data were the mean of three replicates (n = 3). Note: * = results without statistical significance (*p* > 0.05); ** = results with statistical significance (*p* < 0.05).

**Figure 5 ijms-24-07404-f005:**
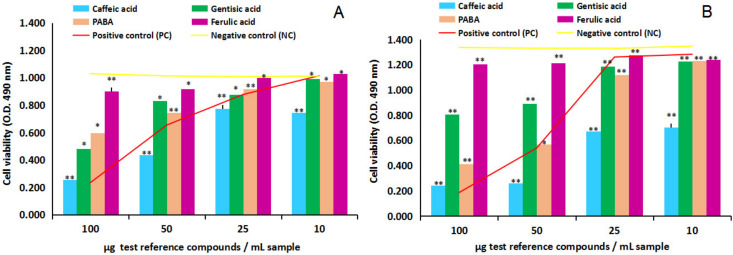
Effect of test reference compounds (caffeic acid, gentisic acid, ferulic acid and para-aminobenzoic acid/PABA) on human tumor brain cell line U87, in anti-proliferative MTS assay. (**A**) U87 cell line viability after 24 h of exposure to test reference compounds. (**B**) U87 cell line viability after 48 h of exposure to test reference compounds. Data were the mean of three replicates (n = 3). Note: * = results without statistical significance (*p* > 0.05); ** = results with statistical significance (*p* < 0.05).

**Figure 6 ijms-24-07404-f006:**
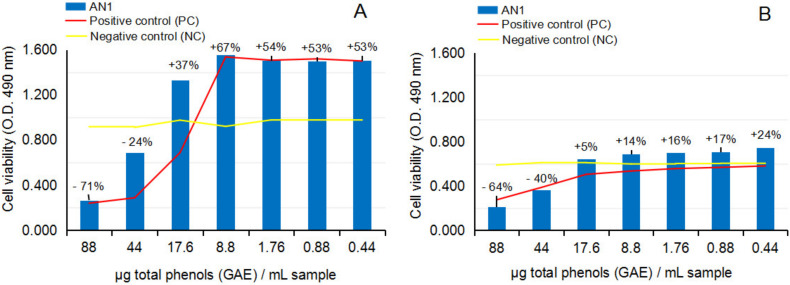
Effect of crude ethanolic extract from fresh *Anemone nemorosa* L. plant material (AN1) on human normal brain cell line NHA and human tumor brain cell line U87, in cytotoxicity MTS assay. (**A**) NHA cell line viability after 24 h of exposure to test vegetal extract AN1. (**B**) U87 cell line viability after 24 h of exposure to test vegetal extract AN1. All determinations were made in triplicate series, and the results are calculated as an average. Graphs A and B also present the percentage of cell viability by comparison to negative control series (CN). According to the statistical calculation, the results of the NHA and U87 cell lines in the cytotoxicity experiment were not statistically significant (*p* > 0.05, n = 3).

**Figure 7 ijms-24-07404-f007:**
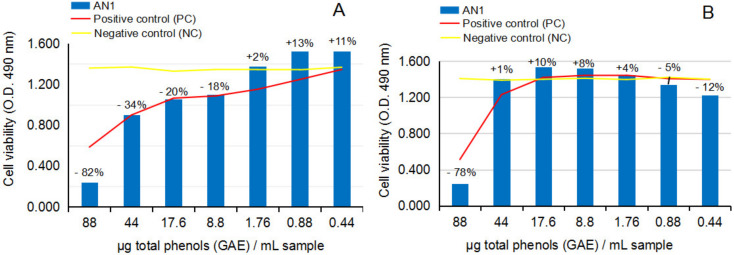
Effect of crude ethanolic extract from fresh *Anemone nemorosa* L. plant material (AN1) on human tumor brain cell line U87, in anti-proliferative MTS assay. (**A**) U87 cell line viability after 24 h of exposure to test vegetal extract AN1; (**B**) U87 cell line viability after 48 h of exposure to test vegetal extract AN1. All determinations were made in triplicate series, and the results are calculated as an average. The graphs present the percentage of the cell viability in series by comparison to positive control sample (CP) series. According to statistical calculation, the results on the U87 cells at 24 h and 48 h were not statistically significant (*p* > 0.05).

**Figure 8 ijms-24-07404-f008:**
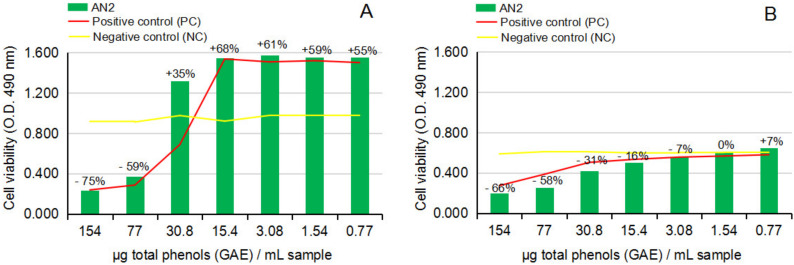
Effect of hot ethanolic extract from dried *Anemone nemorosa* L. plant material (AN2) on human normal brain cell line NHA and human tumor brain cell line U87, in cytotoxicity MTS assay. (**A**) NHA cell line viability after 24 h of exposure to test vegetal extract AN2. (**B**) U87 cell line viability after 24 h of exposure to test vegetal extract AN2. All determinations were made in triplicate series, and the results are calculated as an average. The graphs present the percentage of cell viability in each dilution point series by comparison to negative control point series (CN). By statistical calculation, the results on the NHA cell line were not statistically significant (*p* > 0.05, n = 3), while the results on the U87 cell lines were statistically significant (*p* < 0.05, n = 3).

**Figure 9 ijms-24-07404-f009:**
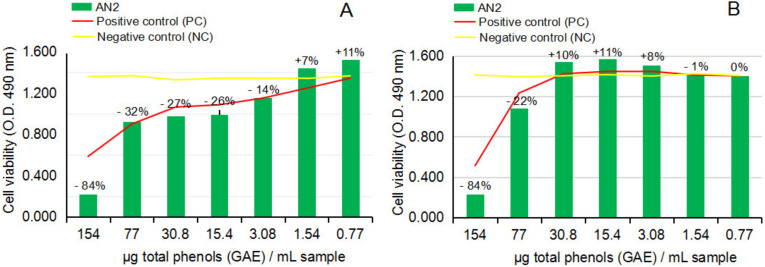
Effect of hot ethanolic extract from dried *Anemone nemorosa* L. plant material (AN2) on human tumor brain cell line U87, in anti-proliferative MTS assay. (**A**) U87 cell line viability after 24 h of exposure to test vegetal extract AN2. (**B**) U87 cell line viability after 48 h exposure to test vegetal extract AN2. All determinations were made in triplicate series, and the results are calculated as an average. The graphs present the percentage of cell viability in each dilution point series by comparison to negative control point series (CN). By statistical calculation, the results on the U87 after 24 h of exposure to AN2 are statistically significant (*p* < 0.05, n = 3), while the results at 48 h were not statistically significant (*p* > 0.05, n = 3).

**Figure 10 ijms-24-07404-f010:**
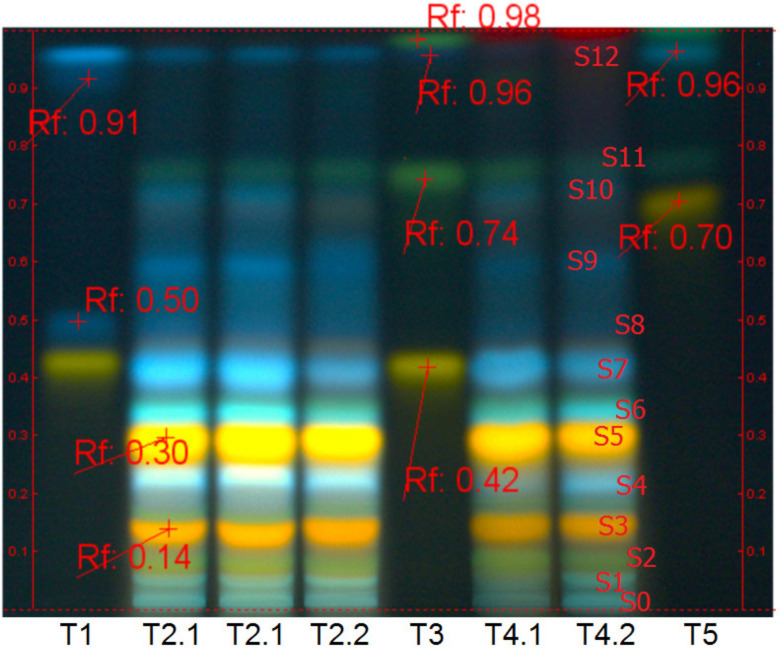
Chemical qualitative analysis (HPTLC method) ofcrude and hot, aqueous and ethanolic extracts from the aerial part of *Anemone nemorosa* L. (*herba et flores*). Solvent system—ethyl acetate: glacial acetic acid:formic acid:water, 100:12:12:26. Identification—NP/PEG No. 28, UV-366 nm. where: Track T1—rutin, chlorogenic acid, gallic acid and caffeic acid (Ref.); Tracks T2.1—hot aqueous extract from fresh *Anemone nemorosa* (duplicate sample); Tracks T2.2—hot aqueous extract dried *Anemone nemorosa* plant material; Track T3—rutin, vitexin, protocatechuic acid and apigenin (Ref.); Track T4.1—AN1 sample <crude/cold ethanolic extract from fresh *Anemone nemorosa* plant material>; Track T4.2—AN2 sample <hot ethanolic extract from dried *Anemone nemorosa* plant material>; Track T5—hyperoside, cosmosiin, rosmarinic acid, and kaempferol (Ref.).

**Figure 11 ijms-24-07404-f011:**
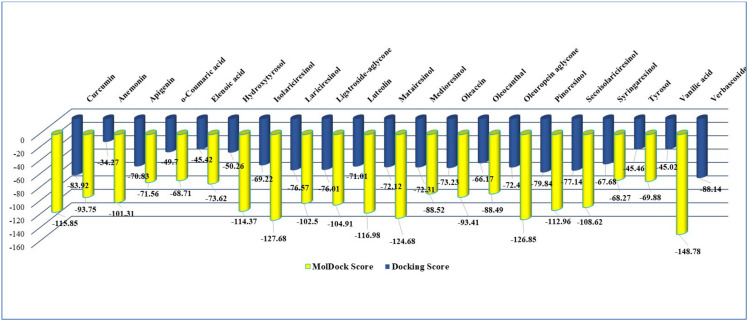
The docking score and MolDock score of the 20 test vegetal compounds in the study in comparison with curcumin.

**Table 1 ijms-24-07404-t001:** The chemical structures and the strength of the interactions between the 20 test molecules docked into 5ZTN (by CLC and MVD software, respectively).

Test Compound	Chemical Structure *	Docking Score CLC	MolDock Score MVD	Ranking Score CLC	Ranking Score MVD
Verbascoside	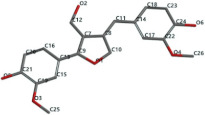	−88.14	−148.78	1	1
Pinoresinol	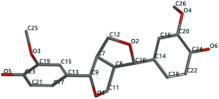	−79.04	−126.85	2	3
Co-crystallized CUR A501, CURCUMIN (5ZTN native ligand)	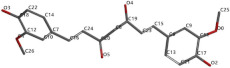	−83.92	−115.85	3	6
Secoisolariciresinol	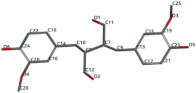	−77.14	−112.96	4	8
Lariciresinol	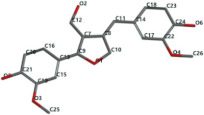	−76.57	−127.28	5	2
Ligstroside aglycone	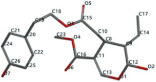	−76.01	−102.50	6	11
Oleuropein aglycone **	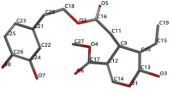	−72.40	−88.49	7	16
Oleacein **	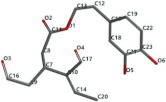	−73.23	−88.52	8	15
Medioresinol	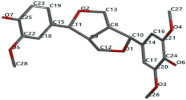	−72.31	−124.68	9	4
Matairesinol	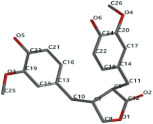	−72.12	−116.98	10	5
Luteolin	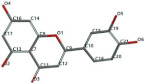	−71.01	−104.91	11	10
Apigenin	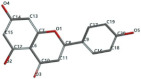	−70.83	−101.31	12	12
Isolariciresinol **	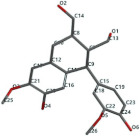	−69.22	−114.37	13	7
Syringaresinol	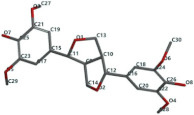	−67.68	−108.62	14	9
Tyrosol	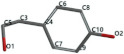	−46.56	−68.27	18	21
Elenolic acid	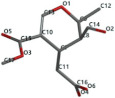	−45.42	−68.71	19	20
Vanillic acid	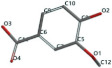	−45.02	−69.88	20	19
*o*-Coumaric acid	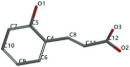	−49.70	−71.56	17	18
Hydroxytyrosol	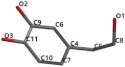	−50.26	−73.62	16	17
Oleocanthal	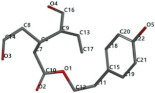	−66.17	−93.41	15	14
Anemonin	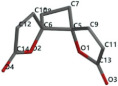	−34.27	−93.75	21	13

* Structures were imported from PubChem database (https://pubchem.ncbi.nlm.nih.gov, accessed on 30 January 2023) [33] and minimized with Spartan software using Molecular Mechanics Force Field (MMFF) [34]; atom labels are arbitrarily established by the software. ** Compounds with the most heterogeneous results in the study.

**Table 2 ijms-24-07404-t002:** Predicted drug-likeness parameters for the 20 + 1 natural compounds found in human diet (alphabetic order).

Test Compounds	No. of Atoms	Weight [Daltons]	Flexible Bonds	Lipinski’s Violations	Hydrogen Donors	Hydrogen Acceptors	logP *
Anemonin	22	192.17	0	0	0	4	0.41
Apigenin	30	270.24	1	0	3	5	4.28
*o*-Coumaric acid	20	164.16	2	0	2	3	1.93
CURCUMIN	49	370.40	9	0	2	6	3.00
Elenoic acid	31	242.23	5	0	1	6	0.12
Hydroxytyrosol	21	154.16	2	0	3	3	0.79
Isolariciresinol	50	360.40	5	0	4	6	2.04
Lariciresinol	50	360.40	6	0	3	6	2.40
Ligstroside-aglycone	48	362.37	8	0	2	7	1.50
Luteolin	31	286.24	1	0	4	6	3.93
Matairesinol	48	358.39	6	0	2	6	3.25
Medioresinol	52	388.41	5	0	2	7	2.25
Oleacein	43	320.34	10	0	2	6	1.11
Oleocanthal	42	304.34	10	0	1	5	1.47
Oleuropein aglycone	49	378.37	8	0	3	8	1.14
Pinoresinol	48	358.39	4	0	2	6	2.28
Secoisolariciresinol	52	362.42	9	0	4	6	2.51
Syringaresinol	56	418.44	6	0	2	8	2.22
Vanillic acid	20	168.15	2	0	2	4	1.26
Verbascoside	80	624.59	11	3	9	15	−0.51
Tyrosol	20	138.16	2	0	2	2	1.14

* logP is the octanol-water partition coefficient calculated by CLC computation.

## Data Availability

Not applicable.

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
