# Peer review of "Potential Benefits of Dietary Plant Compounds on Normal and Tumor Brain Cells in Humans: In Silico and In Vitro Approaches"

_ijms, 2023, doi:10.3390/ijms24087404_

Round 1
Reviewer 1 Report
The study evaluated potential benefits of dietary plant compounds on human normal and tumor brain cells. Both in-silico simulation and in-vitro studies with two cell lines were performed and the manuscript was well-written. It may be published pending some minor revisions.
1) All abbreviations should be introduced at the first time of appearance, such as "DYRK2", "CLC" and "MVD" in Abstract, and "PABA" in Figures 4 and 5.
2) Figure 10 is in a low resolution. Some numbers in Figure 10 are too small to be seen.
3) In Abstract, the authors mentioned that "A. nemorosa ethanolic extracts have potential benefits to neurodegenerative diseases." But which neurodegenerative diseases? By what evidences? In Line 559, Anemone nemorosa ethanolic extracts increased the viability of the cells in culture. However, this does not support the above conclusion.
Author Response
1) All abbreviations should be introduced at the first time of appearance, such as "DYRK2", "CLC" and "MVD" in Abstract, and "PABA" in Figures 4 and 5. - REVISED
2) Figure 10 is in a low resolution. Some numbers in Figure 10 are too small to be seen. - REVISED - Our software version for Camag Linomat 5 is older and we cannot improve the image further; we hope that the replaced chromatogram (Figure 10) allows now a better evaluation of the qualitative aspects of test extracts, the color of the spots and their Rf.
3) In Abstract, the authors mentioned that "A. nemorosa ethanolic extracts have potential benefits to neurodegenerative diseases." But which neurodegenerative diseases? By what evidences? In Line 559, Anemone nemorosa ethanolic extracts increased the viability of the cells in culture. However, this does not support the above conclusion. - REVISED - We agree, the mention “..” in the summary of the article is completely speculative (it was eliminated).

Reviewer 2 Report
The manuscript needs extensive revision.
There are data in the manuscript that do not find description in the methods section. So, there are method and assays used that are not reported in the manuscript.
The figure captions are not complete. Many abbreviations should be written in full.
The method section should be divided into subsections (4.1 - 4.2 - 4.3 etc.).
The in vitro paragraph (line 575) does not report cell culture conditions, which instead should be described in more detail.
The plant extraction part must report and describe the method not just a bibliographic reference.
Figures , excluding those of the in silico method, are of poor quality and are not well described.
Author Response
- The manuscript needs extensive revision. - REVISED
- The article has been revised in its entirety.
- There were rebuild all the diagrams (Figures 4 - 9) and all the comments for the in vitro studies, and there were added new information regarding the statistical significance of the results; the discussions and conclusions sections were reconsidered and the subsections in materials and methods part, therefore highlighting the exactly description of the studies.
- There are data in the manuscript that do not find description in the methods section. So, there are method and assays used that are not reported in the manuscript.- REVISED
- The figure captions are not complete. Many abbreviations should be written in full. - REVISED
- The method section should be divided into subsections (4.1 - 4.2 - 4.3 etc.). - REVISED
- The in vitro paragraph (line 575) does not report cell culture conditions, which instead should be described in more detail. - REVISED
- The plant extraction part must report and describe the method not just a bibliographic reference. - REVISED
- Figures , excluding those of the in silico method, are of poor quality and are not well described. - REVISED

Round 2
Reviewer 2 Report
the manuscript has been revised and needs only minor corrections and typos errors.
Author Response
Esteemed Reviewers, Esteemed Managing Editors of the special issue "Molecular Pathology, Diagnostics, and Therapeutics", IJMS
The authors carefully checked the text once again and made corrections of typos errors and spelling. The authors have added a new paragraph with a better description of the effects of reference substances on the viability of NHA.
We also mention the fact that we do not have an explanation for the numerous words between which the spaces have disappeared; most likely there is an incompatibility between our electronic devices which will be fixed.
Finally, the authors THANK YOU very much for your very careful review of our paper!
Lucia Pirvu et al. April 14, 2023